

# Wi-Fi sensing gesture control algorithm based on semi-supervised generative adversarial network

Chao Wang[1,2,3], Yinfan Ding[1,2,3], Meng Zhou[1,2,3] and Lin Tang[1,2,3]

[1] Joint Laboratory for International Cooperation of the Special Optical Fiber and Advanced Communication, Shanghai University, Shanghai, China
[2] Key Laboratory of the Special Optical Fiber and Optical Access Network, Shanghai University, Shanghai, China
[3] Shanghai Institute of Advanced Communication and Data Sciences, Shanghai University, Shanghai, China

## ABSTRACT

A Wi-Fi-sensing gesture control system for smart homes has been developed based on a theoretical investigation of the Fresnel region sensing model, addressing the need for non-contact gesture control in household environments. The system collects channel state information (CSI) related to gestures from Wi-Fi signals transmitted and received by network cards within a specific area. The collected data undergoes preprocessing to eliminate environmental interference, allowing for the extraction of complete gesture sets. Dynamic feature extraction is then performed, followed by the identification of unknown gestures using pattern recognition techniques. An improved dynamic double threshold gesture interception algorithm is introduced, achieving a gesture interception accuracy of 98.20%. Furthermore, dynamic feature extraction is enhanced using the Gramian Angular Summation Field (GASF) transform, which converts CSI data into GASF graphs for more effective gesture recognition. An enhanced generative adversarial network (GAN) algorithm with an embedded classifier is employed to classify unknown gestures, enabling the simultaneous recognition of multiple gestures. A semi-supervised learning algorithm designed to perform well even with limited labeled data demonstrates high performance in cross-scene gesture recognition. Compared to traditional fully-supervised algorithms like linear discriminant analysis (LDA), Light Gradient Boosting Machine (LightGBM), and support vector machine (SVM), the semi-supervised GAN algorithm achieves an average accuracy of 95.67%, significantly outperforming LDA (58.20%), LightGBM (78.20%), and SVM (75.67%). In conclusion, this novel algorithm maintains an accuracy of over 94% across various scenarios, offering both faster training times and superior accuracy, even with minimal labeled data.

# INTRODUCTION

With the development of information technology (IT) and the broad implementation capability of the Internet of Things (IoT), human gesture recognition operations play a more crucial role in human-computer interaction. In recent years, human-computer

Corresponding author
Lin Tang, bollytom_lin@163.com

interaction research has shifted from manual to non-contact control operations, thus significantly enhancing users' experience. Therefore, non-contact control technology recognizes users' actions to controllable intelligent devices within a certain space. To do so, users issue commands to devices through different gestures, and the key technology is run to make devices recognize commands pertinent to human gestures.

Although conventional methods have good gesture recognition capability, they all have certain disadvantages to a greater or lesser extent. For example, the method based on cameras obtains a user's visual and physical information, which may violate personal privacy (*Gkioxari et al., 2018*). Moreover, extracting and recognizing sequences of human body motions also require a huge amount of computing power. The method based on infrared light cannot recognize gestures in non-line-of-sight conditions. Also, the cost of the device is high and the deployment process is difficult in a larger area. The method based on wearable sensors has a limited perception range and poor user-friendliness (*Guan & Plötz, 2017*; *Yatani & Truong, 2012*). Therefore, all these issues have been pointed out using wireless passive sensing technology based on Wi-Fi signals. Compared to conventional gesture recognition technology, this technology requires no specialized devices, does not result in privacy leaks, and operates effectively in environments affected by external factors such as smoke or darkness.

With the rapid development of wireless sensing technology, non-invasive gesture recognition solutions based on Wi-Fi signals have grown more promising and interesting. The research on non-contact gesture control for smart homes has sparked interest in developing new user interfaces to eliminate the need for conventional control operations. For example, users can control the volume of music while showering or change a song playing on the music system in the living room while cooking by using air gestures. Also, channels can be switched, or a television volume can be adjusted without a remote control while sitting on the couch.

The article proposes a system using cross-scene wireless sensing to control gestures based on a semi-supervised generative adversarial network (GAN) to recognize and realize unknown gestures for smart homes. However, even though the available conventional machine learning algorithms are implemented widely to conduct these kinds of tasks, they have a disadvantage in cross-scene recognition tasks, and their performance becomes poorer. The system generally utilizes "left", "right", "up", "down", and "draw a circle" movements as the default gestures to control smart homes, simulating scenes such as users switching TV channels or adjusting volume without using remote control in daily routines.

The article contributes to

1. develop a human behaviour recognition algorithm based on the Fresnel zone model and specifically construct a reliable microscale gesture detection algorithm,

2. propose a dynamic double threshold gesture interval interception algorithm. A comparison result is presented to show that the proposed algorithm effectively retains the integrity of gesture interception and improves classification accuracy with the conventional Local Outlier Factor (LOF) algorithm and fixed double threshold gesture interval interception algorithm,

3. Propose and design an improved semi-supervised GAN algorithm with an embedded classifier to enhance the classifier's discrimination capability by generating more data. A small amount of labelled data in the source domain and many unlabeled data in the target domain are employed to realize domain-independent gesture recognition.

The rest of the article is outlined as follows: "Preliminary" presents the preliminary. The basic theory of Wi-Fi sensing is presented and discussed in "Basic Theory of Wi-Fi Sensing". "System Design and Implementation" presents the steps of the system design and its implementation. The results and comparison are presented in "Results". The discussion is provided in "Discussion". "Conclusions" concludes the research.

## PRELIMINARY

With the development of wireless sensing technology, many kinds of research have proposed several novel systems based on wireless sensing implementations to detect and recognize human behaviour (*Wu et al., 2017*). In systems related to recognizing human body activities based on wireless sensing, the research methodologies can be split into many categories, such as Frequency Modulated Continuous Wave (FMCW), Channel State Information (CSI), and Radio Frequency Identification (RFID). The FMCW developed by *Wang et al. (2018)* adopted a technology to realize various wireless sensing systems. They proposed a skeleton-based action recognition system. An intermediate representation of a human skeleton in three dimensions was proposed and employed to pinpoint the actions and interactions of multiple people over time. It achieved a precision level similar to a vision-based action recognition system in scenes visible to human eyes.

Nonetheless, when humans are not visible, it can still function accurately. *Li et al. (2019)* proposed BodyCompass, a new sleep posture monitoring system. The BodyCompass implemented a filtered multipath contour feature extractor to estimate radio frequency (RF) reflection signals directly or indirectly emitting from human bodies. Multipath profile features were utilized to estimate the sleeping positions of a person in a particular location by conducting fully connected neural networks composed of multiple layers. The first radio frequency-based unsupervised learning architecture for human perceptual tasks, Trajectory-Guided Unsupervised Learning (TGUL), was introduced (*Yue et al., 2020*). To concentrate on human trajectories using unsupervised learning algorithms and create a data augmentation process appropriate for RF signals, red-green-blue (RGB) data is translated to RF data using cutting-edge unsupervised learning techniques. Numerous empirical findings on various RF datasets demonstrate that the TGUL can reliably enhance the performance of RF-based perceptual models on a supervised learning baseline. *Chang & Lin (2011)* investigate the feasibility of leveraging physiological and behavioral signals extracted from an RF sensing device to characterize metrics indicative of breathing, mobility, and sleep patterns. Results show that many signals significantly distinguish systemic lupus erythematosus (SLE) and healthy participants.

The CSI developed by *Li et al. (2022)* implemented commercial Wi-Fi devices to detect human behavior activities. They proposed two general sensing models: the Fresnel zone and the CSI ratio. *Daqing et al. (2022)* proposed a new smartphone-based respiratory detection system, called Wi-Phone, which can stably monitor human respiration in non-

visual fields (*Wu et al., 2022*). Also, they proposed motion navigation primitive (MNP), a location-independent feature, to address the location correlation issue between human activity and Wi-Fi signals. The MNP achieved over 90% overall recognition accuracy for three distinct gesture sets (*Liu et al., 2021*). *Daqing et al. (2022)* proposed the first independent path single-subject gait recognition system based on commercial Wi-Fi devices, Wi-Path Independent Gait Recognition (Wi-PIGR) (*Gao et al., 2021*). They proposed a series of signal processing technologies to eliminate signal differences caused by walking along any path and generate high-quality path-independent spectra, with an average recognition accuracy of 77.15% by coining the term "domain gap" (*Zhang, Wang & Zhang, 2021*). The gradient map's symbolic mapping was employed as a domain gap eliminator to increase identification accuracy and achieve domain-independent gesture recognition. A three-stage system for pinpointing multiple human activities was suggested, with each stage being designed based on the size of the available data set in the configuration file for the various phases of the system deployment (*Liu et al., 2020*). The proposed single-threshold-based activity extraction algorithm can also pinpoint an activity's start and end points.

Therefore, the progress of wireless sensing technology has enabled breakthroughs in the development of actual application projects in various fields. The article uses the multipath effects of Wi-Fi signals during transmissions to obtain information on human behavior and thus pinpoints distinct gestures. For instance, when people wave their arms, the system senses the signal changes caused by human movements and extracts gesture information from the CSI related to human behaviour for analysis and processing. Finally, a Wi-Fi signal can sense distinct gestures, ultimately achieving the objective of controlling smart homes in indoor environments.

On the other hand, recent research suggests that Fresnel zones are implemented for two distinct aspects of multipath. First, they are determined for the line-of-sight transmission between the satellite and receiver. When the Fresnel zones' boundaries are compared with an obstruction adaptive elevation mask, potentially diffracted signals can be pinpointed and removed from the position prediction procedure. The suggested algorithm increases the percentage of epochs with fixed ambiguities and the precision of the positioning. Second, they investigate the multipath caused by a horizontal and spatially-limited reflector (*Tang et al., 2023*). Passing-object recognition is a fundamental task in intelligent environments but is still challenging since dense multipath propagations occur in standard indoor places. Thus, a new algorithm is proposed based on a Fresnel zone and diffraction algorithms by implementing indoor radio wave propagation attributions to predict the first Fresnel zone maximum and phase difference. The suggested algorithm achieves low missing alarm and direction error rates for single-passing and multipassing scenarios (*Zimmermann et al., 2019*). When outdoor localizations are compared with indoor localizations using the Global Positioning System (GPS), indoors are more complex and hard since radio signals shadow, fade, and reflect. The proposed method aims to localize targets in indoor places based on the Fresnel zones model of radio communication that finds elliptic regions based on targets located using the CSI attained from Commercial Off-The-Shelf Wi-Fi devices (*Shao et al., 2024*).

Before running machine learning algorithms, preprocessing steps such as feature extraction and selection, algorithm selection, and the optimization of the hyperparameters are conducted for sonar data. Instead of using those preprocessing tasks, three distinct mathematical transformations are implemented to cluster underwater objects by employing GADF and GASF transformation algorithms. By doing so, temporal and spatial correlations are considered (*Fei et al., 2020*). The muscle–computer interface implementations try to extract patterns from complex surface electromyography (sEMG) signals; however, enhancing and improving the accuracy of myoelectric pattern recognition is a very challenging operation (*Yousuf et al., 2024*). To do so, the authors suggested a GAF-based 2D representation of the data used in the development of convolutional neural network (CNN)- based classification (GAF-CNN) (*Civrizoglu buz, Demirezen & Yavanoğlu, 2021*). The GAF and GASF transform 1D data into 2D representations, especially time series or sequence data (*Yun et al., 2023*).

The CSI gauges how Wi-Fi signals propagate through an environment. Nevertheless, several scenarios and implementations suffer from insufficient training data since cost, time, or resources are limited. GAN and pre-trained encoders provide efficient and effective human activity recognition using Wi-Fi data (*Fan, Wen & Lai, 2023*). A rich information set can be generated in indoor environments, such as smart homes, by focusing on occupant numbers and their movements. The proposed model is compared with other algorithms when GAN uses data augmentation and no data augmentation. The results suggest that GAN-based CNN has better outcomes (*Tian et al., 2024*).

## BASIC THEORY OF WI-FI SENSING

### Wireless signal propagation model based on CSI

The CSI carries information about Wi-Fi signals travelling through multiple paths from a transmitter to a receiver. The received CSI signal is a superposition of all path components, including the line-of-sight path and various reflected paths caused by the surrounding environment, such as walls and furniture, as shown in Fig. 1. In gesture recognition tasks, reflex paths can be further split into environmental and hand reflex paths. The hand reflex path is dynamic because the hand is a moving entity. Therefore, a CSI signal consisting of static and dynamic components is expressed by

$$H(f, t) = H_s(f, t) + H_d(f, t) = H_s + A(f, t)e^{(-j2\pi(d(t))/\lambda)} \tag{1}$$

where $H_s$ and $H_d$ represent static and dynamic phasor components, $A(f, t)$ and $e^{(-j2\pi(d(t))/\lambda)}$ represent attenuation and phase shift, respectively, and $\lambda$ denotes the wavelength. In an actual wireless communication link between the transmitting antenna and the receiving antenna of the Wi-Fi system, the CSI obtained from the transmitted packets represents an $N_T \times N_R \times N_C$ complex matrix, where $N_T$, $N_R$, and $N_C$ denote the number of transmitting antennas, the number of receiving antennas, the number of orthogonal frequency-division multiplexing (OFDM) subcarriers. Because the CSI exists in complex form, the amplitude and phase scores of the CSI can be obtained.

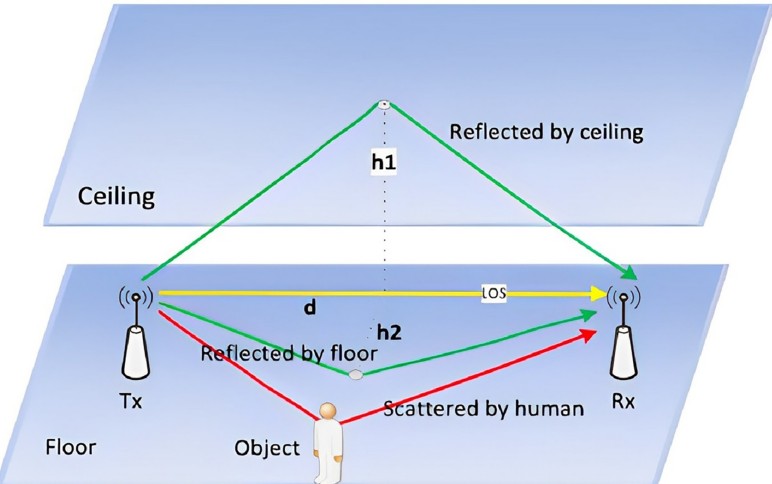

**Figure 1 Wireless signal propagation model.** In Wi-Fi awareness, the channel status information carries the information that the Wi-Fi signal travels through multiple paths from the transmitter to the receiver. The received CSI signal is a superposition of all path components, including the line of sight path and various reflection paths caused by the surrounding environment (such as walls and furniture).

## Fresnel zone sensing model

It is important to theoretically understand the intrinsic nature of changes in CSI signals caused by hand movements. The Fresnel zone theory was introduced into a work related to CSI in a line-of-sight environment, which marks the phase of the gradual transition from a pattern-based to a model-based detection system by utilizing a Wi-Fi signal.

In the field of wireless communication and perception, the Fresnel region refers to the confocal ellipsoid sequence of two focal points corresponding to the transmitting and receiving antennas whose geometries are determined by the wavelength of the wireless signal and the distance between the receiving and transmitting antennas. Given the wavelength $\lambda$ of a wireless signal, the transmitting and receiving antennas are represented by $T_x$ and $R_x$, respectively, a Fresnel region containing $n$ confocal ellipsoids can be constructed by

$$|T_xQ_n| \ + \ |Q_nR_x| - |T_xR_x| = \ n\lambda/2 \qquad (2)$$

where $Q_n$ represents a point on the n-th ellipsoid, the innermost ellipse is defined as the first Fresnel region, the ellipse ring between the first and second ellipses is defined as the second Fresnel one, and the extension delineates the n-th Fresnel one. In contrast, each ellipse is defined as the boundary of the corresponding Fresnel region.

The basic principle of wireless sensing is to analyze the scattering path of a human body by employing the direct path signal and the reflected signal of the ground and wall. By examining the diffraction and reflection models existing in the Fresnel zones formed between transmitting and receiving signals, the two models have different proportions in detecting human movements and behaviors in different Fresnel zones. The diffraction model is dominant when a human target is in the first Fresnel region. On the other hand, the reflection model dominates the outside of the first Fresnel region. Given the lack of

reliable micro-scale detection models, a human behavior recognition model based on the Fresnel zone was constructed to provide specific model construction and theoretical guidance for experimental research (*Yatani & Truong, 2012*).

For the motion of any gesture, as an object passes through a series of Fresnel regions, the received signal shows a continuous sine-wave-like waveform. If the length of a reflected path caused by a moving object changes by less than one wavelength, the received signal is only a fragment of a sinusoidal waveform. For example, the fingers move within the sensing range of a pair of Wi-Fi transceivers. As a finger moves, the length of the reflection path changes, and the dynamic phasor component rotates accordingly. If the length of the reflection path changes by less than one wavelength, the superimposed CSI changes along the arc. The direction of the rotation of the dynamic component is tightly coupled with the direction of the finger movement, leading to an increase or decrease in the length of the reflection path.

## SYSTEM DESIGN AND IMPLEMENTATION

### A system framework

The proposed algorithm implements the CSI carried by wireless signals to recognize gesture actions. It can be split into four modules, as shown in Fig. 2.

(1) Data collection: two laptops equipped with Intel 5,300 wireless network cards are used as experimental devices. One laptop functioned as a transmitter and the other as a receiver, collecting the CSI related to gestures within a particular area.

(2) Data preprocessing and gesture segmentation: The collected raw CSI is filtered by low-pass filters and its dimension is reduced by running the principal component analysis (PCA). The double threshold gesture segmentation algorithm is then employed to segment the data containing gestures, and the waveform data of the segmented gestures is trained.

(3) Dynamic feature extraction: The CSI signals are converted into the Gramian Angular Summation Field (GASF) graphs to process the data. Compared with conventional time-domain feature extraction, enough dynamic action information in the CSI is retained.

(4) Model training and evaluation: An improved semi-supervised generative adversarial network (GAN) was designed, the generator was optimized, a lightweight convolutional neural network was implemented as a classifier to assess the performance, and the conventional machine learning algorithms were implemented to compare the results.

### Materials and methods

#### Data collection

The sender used a PC equipped with an Intel 5,300 wireless network card to send Wi-Fi signals, and the receiver collected the CSI data, including gestures from another PC equipped with an Intel 5,300 wireless network card through the CSI Tool (*Feng et al., 2019*). The height of the antenna from the ground is 1 m, and the distance between the antennas is about 5 m. The tested person sits in the middle area of the two computers. In the detection area, the tested person performs five kinds of wave actions, namely, "left", "right", "up", "down", and "draw a circle", respectively, and collects 10 s in 4 scenes, in

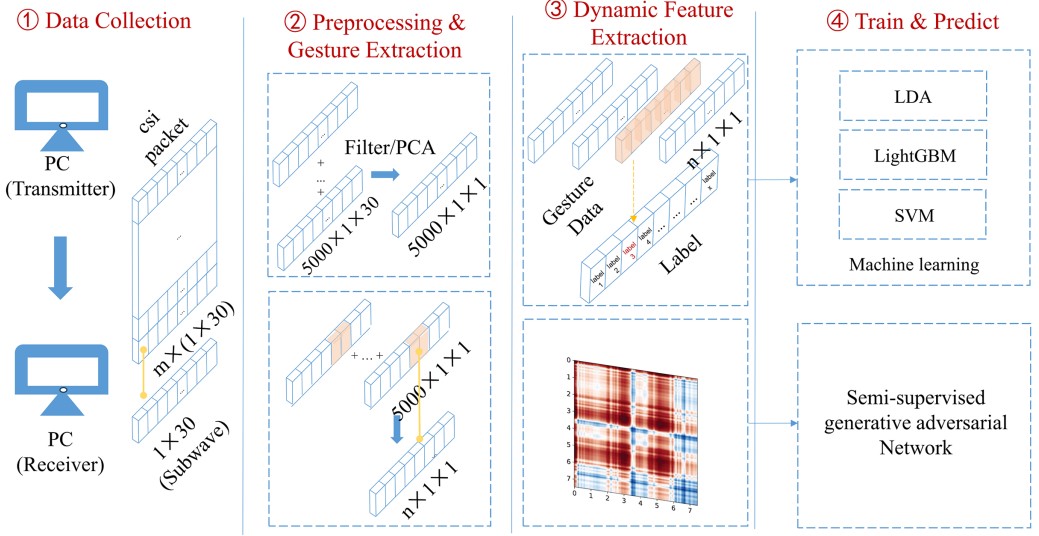

**Figure 2 Schematic diagram of overall system architecture.** The gesture recognition algorithm proposed in this article uses the channel state information carried by wireless signals to recognize gesture actions. It can be divided into four modules as shown here.

which the gesture action is about 2 s. Each action is repeated 100 times as a data set. The working frequency band of the router can be 2.4 and 5 GHz, respectively. Considering that many devices utilize the 2.4 GHz band and the noise interference is too large, the 5G working frequency band is selected in this experiment.

The article presents a wireless signal sensing system that detects a Wi-Fi signal's channel bandwidth of 20 MHz. The system uses two subcarriers as its basic group unit and obtains information about the channel state of the 30 channels from each packet.

## Data preprocessing and gesture segmentation

### Preprocessing of raw CSI data

The CSI scores provided in commercial Wi-Fi network cards contain inherent noise. In real life, gesture speed is generally no more than 1 m/s. Therefore, for Wi-Fi equipment working at 5 GHz, the frequency of the CSI's amplitude waveform change caused by human behavior is $f = 2v/\lambda \approx 40$ Hz. The pertinent data is predominantly found at the lower frequency range of the spectrum, with noise typically occurring at higher frequencies. In such scenarios, the Butterworth low-pass filter is considered more suitable for noise reduction than alternative filters. This filter is advantageous as it minimally distorts the phase information within the signal and provides a maximally flat amplitude response in the passband, thereby avoiding the excessive distortion of a gesture motion signal (*Halperin et al., 2011*). Consequently, this research employs the Butterworth low-pass filter to denoise the original CSI data initially.

The time series form of the CSI in each different subcarrier caused by human behavior is correlated when each transmission-reception antenna pair is considered. The PCA adopted by the system takes advantage of the correlation to perform secondary denoising

action, removing uncorrelated noise components that conventional low-pass filters cannot remove in the signal, which results in pure CSI time series data and improves the accuracy of gesture recognitions. The final data has a reduced dimension and less computational complexity. Thus, the first principal component is selected as the preprocessed CSI data due to having the main and consistent power changes caused by the target motion (*Qian et al., 2017*).

## Dynamic double threshold gesture segmentation algorithm

The preprocessed CSI waveform comprises an action interval and no action interval. If any machine learning classification algorithm is directly applied to the filtered CSI waveform, the classification accuracy will be lower. Therefore, it is necessary to partition the CSI action interval accurately, including gestures.

The conventional gesture interception algorithm adopts the local outlier factor (LOF) algorithm (*Qian et al., 2017*). If the LOF score is less than 1, it indicates that the density of the point is higher than that of its neighbouring points, which is called dense. When the score is greater than 1, the density of the point is less than that of its neighbourhood points. Hence, the point is more likely to be an anomaly. The score of the LOF can be obtained according to the calculation steps, as shown in Fig. 3. The section in the red box is not the starting point of the gesture, according to Fig. 4. Still, its LOF score is significant, resulting in a misjudgment of the starting point of the gesture so that the gesture cannot be accurately intercepted.

The variance of the CSI without activity is significantly more stable than that with activity. If the CSI waveform fluctuates beyond this threshold, it indicates activity. The article utilizes an enhanced dynamic double threshold gesture interception algorithm to achieve gesture interval interception. In comparison to conventional gesture segmentation algorithms such as the LOF outlier detection algorithm and fixed double threshold gesture interception algorithm (*Liu et al., 2020*), the proposed algorithm, called the dynamic double threshold gesture interception algorithm, utilizes a dynamically adjustable sliding window for detecting the start and end points of activities, thereby achieving more precise gesture interception.

The idea of the gesture segmentation algorithm is to set two sliding windows, W and w, with different sizes, corresponding to two thresholds, h and u, according to the difference in the average absolute deviation between the gesture interval and the non-gesture interval. Using the sliding window to traverse the CSI sequence, the average absolute deviation score $W_L$ of the left window in the ith point is calculated, as shown in Eq. (3). The average absolute deviation score $W_R$ of the right window is computed, as shown in Eq. (4). When $W_L < h$ and $W_R > h$, the ith point is judged to be the start point. To improve the accuracy of segmenting the gesture action interval, the average absolute deviation score $w_r$ of the right small window w in the ith point is also calculated, as shown in Eq. (5). When $W_L < h$, $W_R > h$, and $w_r > u$ are all met, the ith point is judged to be the starting point. Similarly, when the ith point's $W_L > h$, $W_R < h$, and $w_r < u$, the point is judged to be the endpoint. In this way, through the dual-threshold judgment rule before and after, it is possible to

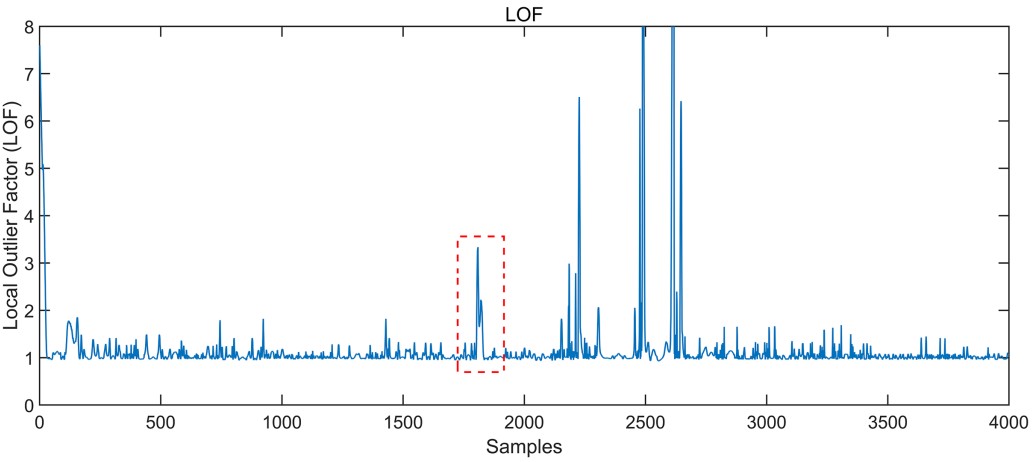

**Figure 3 Local outlier factor LOF of the signal.** The value of local outlier factor LOF can be obtained according to the calculation.

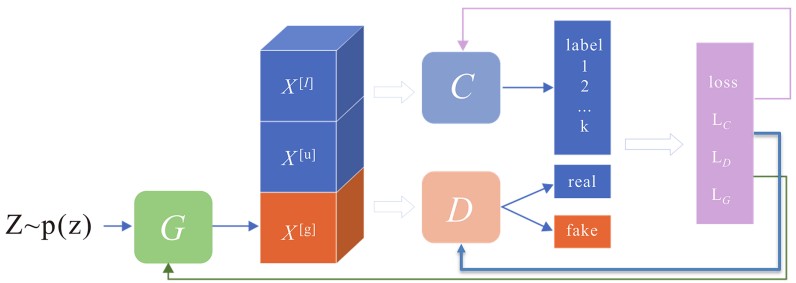

**Figure 4 Semi-supervised generative adversarial network architecture.** The network structure consists of three main parts: classifier C, generator G, and discriminator D.

eliminate non-action interval interference as much as possible so that the segmented interval contains as much action information as possible.

$$W_L = \frac{1}{W} \sum_{j=i-W+1}^{i} |H(j) - \bar{H}(i-W+1:i)| \tag{3}$$

$$W_R = \frac{1}{W} \sum_{j=i}^{i+W-1} |H(j) - \bar{H}(i:i+W-1)| \tag{4}$$

$$w_r = \frac{1}{w} \sum_{j=i}^{i+w} |H(j) - \bar{H}(i:i+w)| \tag{5}$$

The window size is set based on the duration of each action during data collection. The size of the sliding window W is assigned to 300 samples, and the size of the small window w is 1/4 of the large window.

Since the variance of the Wi-Fi CSI is much more stable when there is no activity than when the activity exists, the available works generally employ fixed thresholds to detect an activity's start and end points. If the CSI waveform fluctuates beyond this threshold, the

activity is considered to have occurred. However, in a real-time system, the environment changes over time and can cause wild fluctuations in the variance of the Wi-Fi CSI even when no activity occurs. Therefore, fixed thresholds are not suitable for real-time systems.

The proposed dynamic double threshold gesture interception algorithm can dynamically tune the threshold score, employing the weighted mean of the average absolute deviation scores of all windows and the threshold score of k minimum average absolute deviation scores represented by

$$T1 = \omega V(i) + (1 - \omega) \sum_{j \in N_{low}} V(j) \tag{6}$$

where $\omega$ represents the weight of two average absolute deviation scores, N and $N_{low}$ represent the set of all and k minimum absolute deviation scores, respectively. Because the environment may change over time, the threshold is updated every 2 min to split the activity accurately. Specifically, assuming that the current threshold is denoted by $T1_{cur}$, a new threshold $T1_{new}$ is calculated by implementing the newly collected CSI data. These two thresholds are then combined to generate a new activity segmentation threshold score.

$$T1 = \alpha T1_{cur} + (1 - \alpha) T1_{new} \tag{7}$$

where $\alpha$ represents the parameter that adjusts the weight of the two parts, in this way, thresholds can be updated based on new data and become more suitable for use in new environments. After finding the start and endpoints, a small window is set at the current moment, and the average absolute deviation in the window is computed and compared with the threshold $u$ to ensure that the intercepted interval includes the action interval as much as possible. The point is the start point or the endpoint.

## Dynamic feature extraction

### Feature extraction in the conventional time domain

A feature represents some prominent properties of a sample and is the key to distinguishing gestures. It can be determined whether a sample belongs to a certain class by judging the features extracted from it when classifying or recognizing the sample. After data preprocessing and gesture segmentation, features were extracted from a CSI amplitude as feature vectors for gesture recognition. The manuscript selects eight time-domain characteristics including standard deviation, motion period, median absolute deviation, quartile distance, information entropy, maximum, minimum, and difference score.

Although features can reflect the information of a sample's properties, different features play distinct roles in classification, and some features may also be redundant, such as the association among the maximum, minimum, and range. Feature selection evaluates the size of the information of different features by a feature evaluation index, removing redundant and negative features from the feature set. The feature selection process reduces the number of attributes but improves the classification performance. After the feature selection process, the classification model also has better generalization and robustness and saves a lot of computational resources and time.

The manuscript adopts the Sequential Forward Selection (SFS) algorithm and Sequential Backward Selection (SBS) algorithm to efficiently screen the subsets of the candidate optimal feature parameters, improving the generalization capability of the classification models.

The SFS algorithm starts from an empty set and adds features $X^+$ sequentially so that when combined with the already selected features $Y_k$, the target function $J(Y_k + X^+)$ is optimal.

The SBS algorithm begins with the complete set and removes features $X^-$ sequentially so that when combined with the already selected features $Y_k$, the target function $J(Y_k - X^-)$ is optimal.

### From CSI to GASF diagram

Conventional time-domain features will lose a lot of signal information so that CSI signals can be converted into GAF graphs for subsequent signal processing (*Boukhechba et al., 2023*). The Gramian Angle Field (GAF) method is composed of two different types: the Gramian Angle Sum Field (GASF) and the Gramian Angle Difference Field (GADF). The GASF and GADF images are obtained by computing the cosine function of two angle sums and the sine function of two angle differences, respectively. The GASF image retains more CSI information. Before converting the CSI to a polar coordinate system, it is necessary to scale the CSI to the (0 1) interval employing min-max normalization.

Since the GASF is computed by employing the Angle $\theta$ in the polar diagram, $H_i$ should be represented by the corresponding polar coordinate point. The corresponding angle is computed by

$$\theta_i = \arccos(\hat{H}_i), 0 \leq \hat{H}_i \leq 1. \tag{8}$$

Therefore, the CSI is converted into images for subsequent processing by utilizing the GASF.

## Model training and testing

The key to implementing gesture recognition lies in selecting the classification algorithm. In the selection process, the complexity and performance of an algorithm need to be considered. Therefore, the manuscript adopted LDA, LightGBM based on a decision tree, and another high-performance SVM classification algorithm to train the feature matrix of the CSI since their complexity levels are low. Finally, the performance of the three classification models was compared.

### Machine learning classification algorithms

Linear discriminant analysis (LDA) recognizes gesture actions (*Breunig et al., 2000*) as a typical pattern recognition method belonging to a group of linear classifiers. The core idea is to map multiple categories of data sets to an optimal discriminant vector space to extract category information and effectively compress the spatial dimension of features. This method minimizes the distance between classes, which means that the separability of the

model is the best in the whole subdomain. This criterion is also known as Fisher's differential treatment principle.

The Light Gradient Boosting Machine (LightGBM) employing gradient-boosted decision trees expedites training duration and diminishes memory consumption (*Guo, Xu & Chen, 2019*). Thus, the performance is enhanced. The LightGBM employs a series of decision trees for sample classification, wherein the gradient of the preceding decision tree is addressed and reduced by the subsequent decision tree until the residual reaches a minimal threshold, thereby achieving an optimal solution. The classification process of the LightGBM is delineated as follows: Let $F_m$ be the initial model, $h_m$ be the base estimator, and $Y = F$ is predicted by minimizing the predefined loss function $L(Y, F)$, where $Y$ and $F$ denote the expected output function and the predicted output function, respectively. The initial model $F_m$ is fitted to the basis estimator $h_m$ so that $F_m + h_m = Y$ and residual $h_m = Y - F_m$ is calculated. The residual gradient of the next subsequent estimator $h_{m+1}$ and the previous estimator is fitted so that $F_{m+1} = F_m + h_{m+1}$ and a new model $F_{m+1}$ is constructed. Iterations are continued until the loss function $L(Y, F)$ is minimized and the optimal $Y = F$ is reached. The LightGBM selectively focuses on data instances by exhibiting significant gradients to estimate the information gain. This approach attains precise information gain by using a limited dataset, thereby diminishing time and memory complexities.

The support vector machine (SVM) is run utilizing the LIBSVM software package developed by *Ke et al. (2017)* for classification. The LIBSVM is recognized for its simplicity, speed, and efficiency in pattern recognition and regression tasks. The software offers a broad range of default parameters and cross-validation functions for SVM, minimizing the need for extensive parameter tuning.

The LIBSVM selects the radial basis function (RBF) as a kernel function, and its decision function is delineated by

$$\text{Predict}_{\text{label}} = \text{sgn}\left(\sum_{i=1}^{n} w_i e^{-\text{gamma}||x_i - x||^2} + b\right) \tag{9}$$

where $e^{(-\text{gamma}||x_i - x||^2)}$ represents the kernel function used by the SVM, $x_i$ denotes the support vector in the training dataset, $x$ represents the support vector sample of the predicted label, $w_i$ shows the coefficient of the support vector in the decision function, $b$ denotes the opposite of the constant term in the decision function, and $\text{Predict}_{\text{label}}$ represents the decision result. Hence, the system must ascertain the parameters of the decision function to deduce the label of the sample data under evaluation, thereby constructing the current action status. The precision of the system's recognition is contingent upon the selection of decision function parameters, thereby giving rise to the need for parameter optimization.

### Semi-supervised adversarial network generation

Considering the complexity and difficulty of collecting a large amount of CSI data, achieving a high gesture recognition accuracy is particularly significant, even when only a small amount of labelled data exists. Therefore, a semi-supervised GAN is proposed to train gesture classification. The network has the following characteristics: First, it can use

many unlabeled data to extract gesture features and improve classification accuracy. Second, it can generate large data through the generation network to enhance the classification model's generalization capability.

The current methods usually carry out semi-supervised training by adjusting the discriminator structure. However, these methods do not fully consider the potential improvement of the generated data on the classification performance and neglect to enhance the generation performance when the improvement of the model's fitting capability is considered. Thus, the article proposes an improved semi-supervised GAN based on the Margin-GAN heuristic.

The network structure consists of three main sections: classifier C, generator G, and discriminator D, as shown in Fig. 4. The discriminator serves the same purpose as in a standard GAN, distinguishing whether the sample is from an accurate distribution or produced by a generator. The classifier is trained to increase the classification boundaries of actual samples (both labelled and unlabeled) while simultaneously decreasing the boundaries of the generated fake samples. The generator aims to produce counterfeit samples that look realistic and have significant boundaries, designed to fool both the discriminator and the classifier.

The generator contains two fully connected blocks and four convolution blocks to generate high-quality data. Each convolutional block consists of one deconvolution layer, one batch normalization layer (BNL), and one activation layer. The first three activation functions are modified linear units (ReLU), and the final output activation functions and Tanh functions, respectively. The convolution kernel size is assigned to $4 \times 4$, the step size is set to 2, and the input is randomly generated noise.

The article constructs a CNN classifier with four convolution layers to extract multi-scale GAF image features. The classifier consists of four convolution blocks, each followed by a batch normalization layer (BNL), which improves the convergence speed. The first two convolution blocks include one convolution layer, 1 BNL, one activation layer, and one maximum pooling layer, and the last convolution block contains only the convolution layer and the activation layer. The convolution kernel size is assigned to $4 \times 4$, the step size is set to 1, and the pooling layer size is set to 2, activated using the Leaky ReLU function. A dropout layer is added after the third convolutional block to reduce overfitting, and the classification results are output through the softmax function.

The discriminator consists of two convolution blocks and two fully connected layers. Each convolution block contains a convolution layer, a BNL, and an activation layer. The convolution kernel size is assigned to $4 \times 4$, the step size is set to 2, and the activation function is also used as the Leaky ReLU function.

## RESULTS

To illustrate a "left" gesture, the overall gesture interception and dynamic feature extraction process are shown in Figs. 5–8.

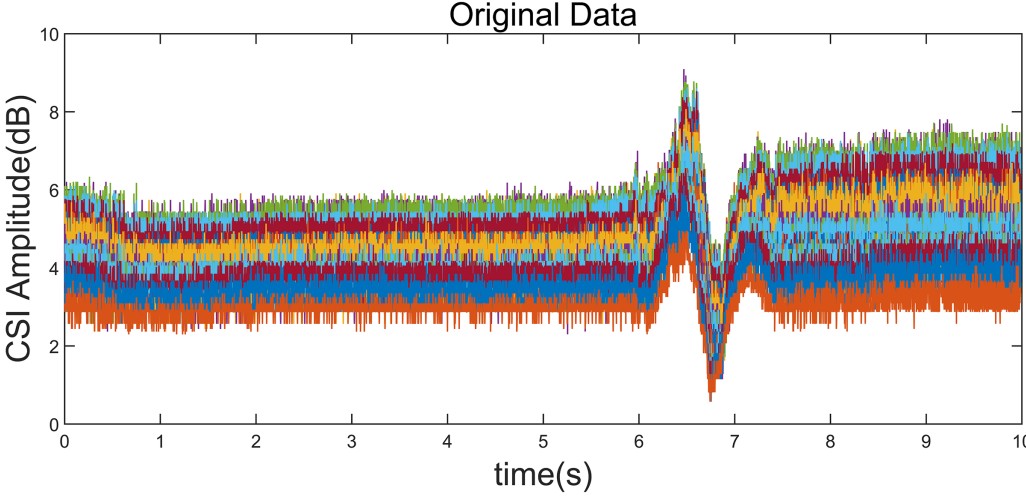

**Figure 5 Original CSI amplitude curve.** Taking the "left" gesture as an example, the overall gesture interception and dynamic feature extraction process is shown.

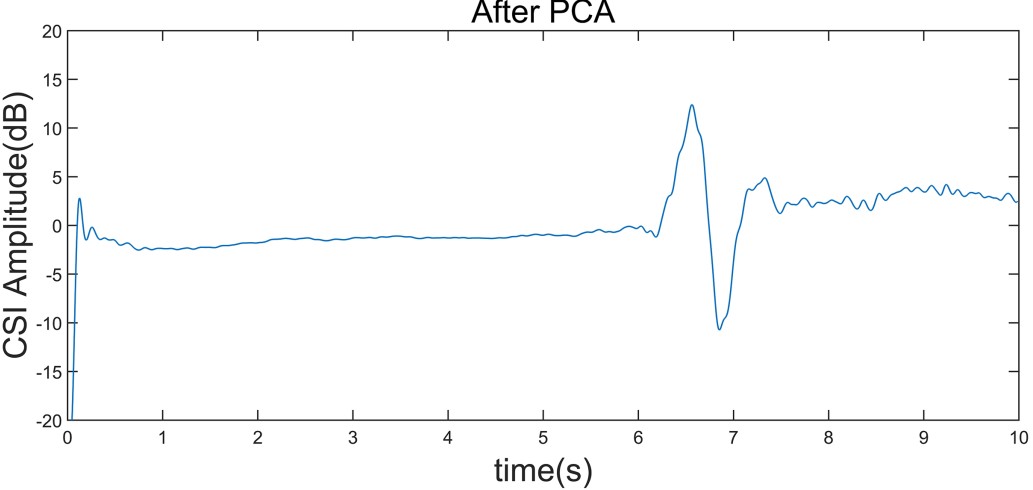

**Figure 6 CSI amplitude curve after gesture interception.** Taking the "left" gesture as an example, the overall gesture interception and dynamic feature extraction process is shown.

## Overall performance

The 100 data containing each type of gesture were divided into the training and test sets with a ratio of 4:1. Table 1 lists the mean recognition accuracy of the proposed system with distinct classification algorithms in the same scenario and across scenarios, respectively; the histogram is shown in Fig. 9. The average accuracies of the LDA, the LightGBM, and the SVM can reach 82.40% and 58.20%, 93.20% and 78.20%, and 94.67% and 75.67%, respectively. On the other hand, the mean accuracy of gestures in the improved semi-supervised GAN can reach 98.33% and 94.67%. The results designate that the recognition accuracy of conventional machine learning algorithms decreases sharply across, while the

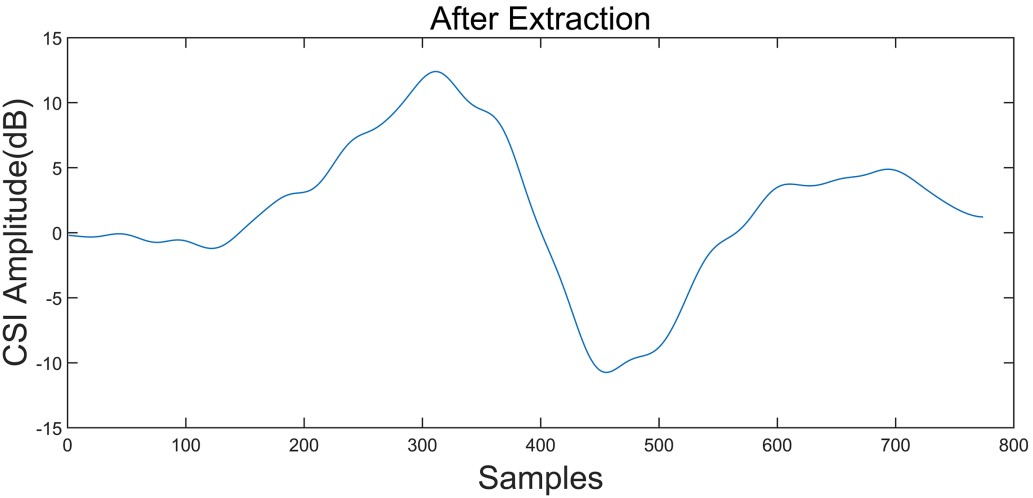

**Figure 7 Local outlier factor LOF of the signal.** Taking the "left" gesture as an example, the overall gesture interception and dynamic feature extraction process is shown.

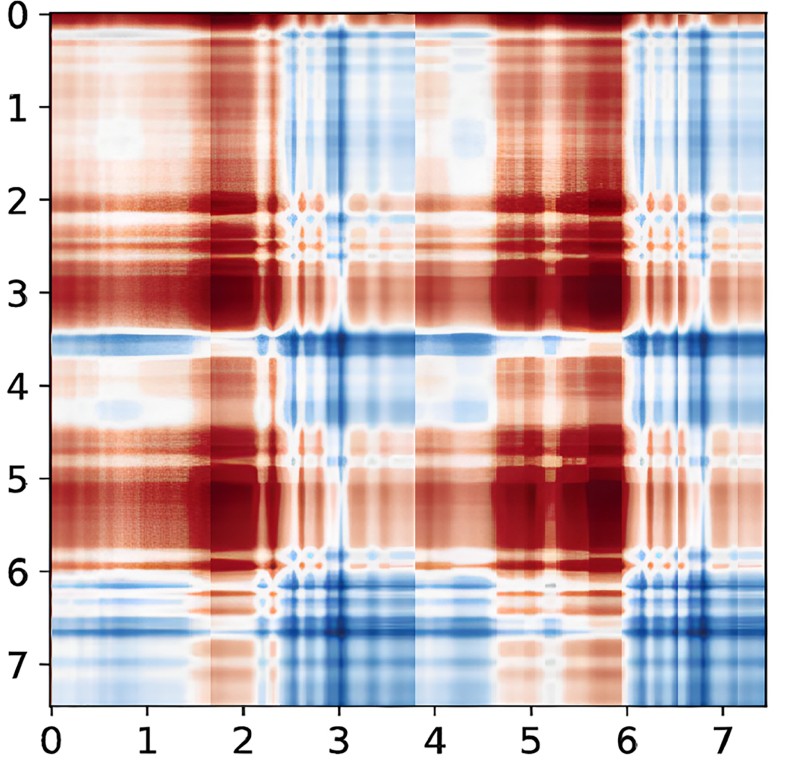

**Figure 8 CSI signal to GASF diagram.** Taking the "left" gesture as an example, the overall gesture interception and dynamic feature extraction process is shown.

Table 1  System recognition accuracy of different classification algorithms in the same scenario and across scenarios.

|  | LDA | LightGBM | SVM | Our |
|---|---|---|---|---|
| Same scene | 82.40% | 93.20% | 94.67% | 98.89% |
| Cross-domain | 58.20% | 78.20% | 75.67% | 95.67% |

**Note:**
The results show that the recognition accuracy performance of traditional machine learning in the case of cross-scene decreases sharply, while the algorithm proposed in this article can achieve a high cross-validation accuracy in the same scene and across scenes, and achieve the expected effect of fast training models with small samples.

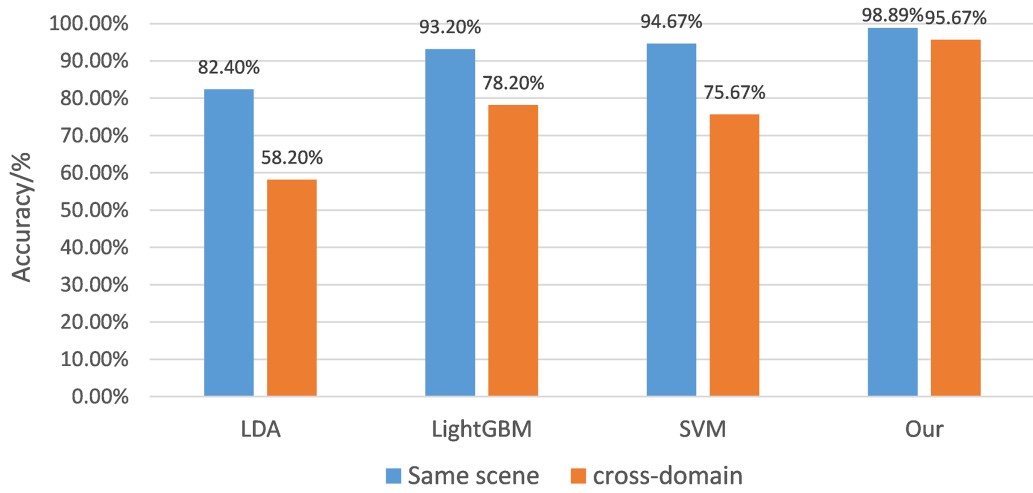

Figure 9  **Recognition accuracy histogram of different classification algorithms.** The 100 data of each type of gesture were divided into the training set and the test set according to the ratio of 4:1. Table 1 lists the average recognition accuracy of the system corresponding to different classification algorithms in the same scenario and across scenarios, and the corresponding data histogram is shown.

proposed algorithm achieves a high cross-validation accuracy in the same scene and across scenes and also achieves a fast training stage with small samples.

# DISCUSSION

## Performance evaluation

### Performance analysis of different gesture interception algorithms

The article uses the LOF outlier detection algorithm, fixed double-threshold gesture interception algorithm, and dynamic double-threshold gesture interception algorithm to intercept and evaluate gesture movements. The accuracy of gesture interception is shown in Fig. 10.

The evaluation criterion of recognition accuracy was the number of accurately intercepted gestures divided by the total number of gestures.

Figure 10 depicts the accuracies of the LOF algorithm, fixed double threshold interception algorithm, and dynamic double threshold interception algorithm, whose scores are 81.40%, 94.10%, and 98.2%, respectively. The accuracy of the dynamic double threshold interception algorithm exceeds the other two algorithms, which indicates that

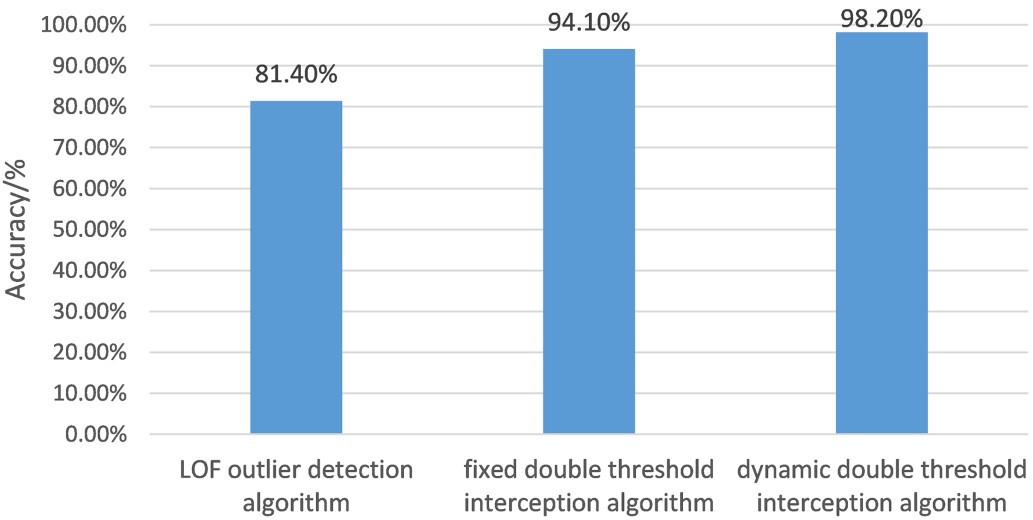

**Figure 10** **The histogram of gesture interception accuracy of different algorithms.** In this article, the LOF outlier detection algorithm, fixed double threshold gesture interception algorithm and dynamic double threshold gesture interception algorithm are respectively used to intercept gesture movements and evaluate gesture movements after interception. The accuracy of gesture interception is shown.

the algorithm proposed in the article makes the gesture recognition effect more accurate and thoroughly verifies its feasibility.

### The analysis of cross-domain performance

The data of one environment is regarded as labelled data for different environments, and the data of the other three environments are randomly divided into test sets and unlabeled data with a ratio of 1:4. The test results are shown in Fig. 11. The proposed algorithm can achieve an average accuracy of 95.67%. Environment 1 achieved the highest accuracy rate of 96.67%. Even the most complex environment 3 had 94.51% accuracy. The implemented label data comprises only 1/3 of the total data implemented in the supervised algorithm. Still, the accuracy of the supervised algorithm is 96.10%, which is consistent with the performance of the proposed algorithm.

The experiments suggest that the proposed algorithm can achieve high gesture classification accuracy even with a small amount of labelled data and a large amount of unlabeled data. The sampling results of the generated data by the generator show that the generated samples cover a variety of gestures, which can effectively represent distinct characteristics of gestures, which is conducive to the training of classification algorithm, and is suitable for scenarios whose characteristics are cross-domain and small sample and needs a fast training stage.

### Performance and complexity analysis of the different machine learning algorithms

Table 1 suggests that the performance of the SVM is much better than the LDA. The main reason is that the structure of the LDA is relatively simple, and linear classification is achieved by projecting high-dimensional samples onto the low-dimensional plane. The

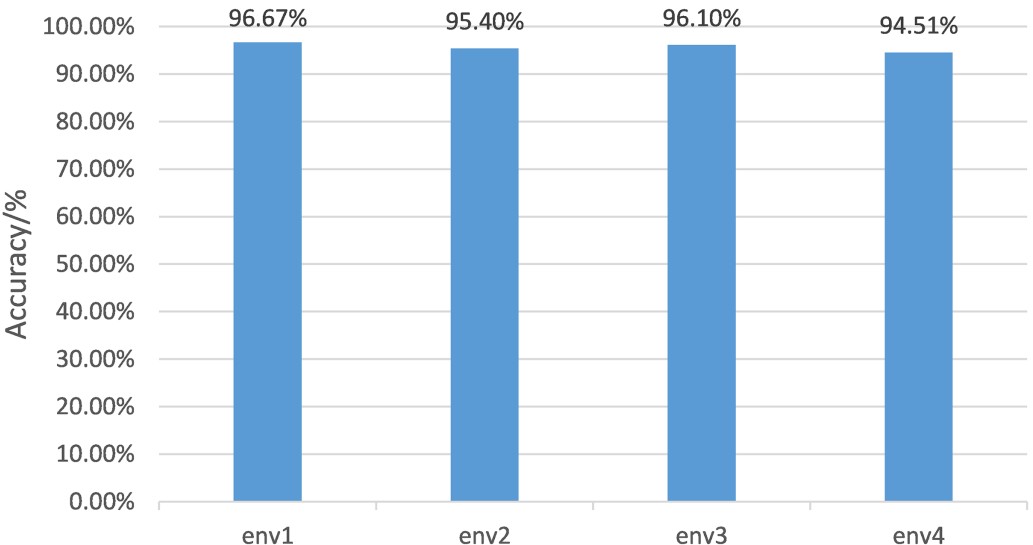

**Figure 11 Semi-supervised gesture recognition accuracy in different environments.** For different environments, the data of one environment is regarded as labeled data, and the data of the other three environments are randomly divided into test sets and unlabeled data in a ratio of 1:4. The test results are shown here.                                                 

training operation is fast but has a poor performance. The algorithm complexity is represented by $O(d^2 * n)$, where $d$ and $n$ represent the sample dimension and the number of samples, respectively. The SVM maps discrete hyperplanes to a surface based on kernel function, which can effectively overcome the problems of conventional algorithms in the learning process. The trained model has good performance and high recognition accuracy, but the training speed is slower than the LDA. The algorithm complexity is denoted by $O(d^2 * n^2)$, where $d$ and $n$ represent the sample dimension and the number of samples, respectively. In future scenarios, the SVM will be more reliable and practical, sacrificing a small amount of training time, but can be exchanged for better performance.

Moreover, the optimization model can be further trained during the non-use period without affecting the user experience. The mean gesture accuracy of the LightGBM can also reach 92.67% and 95.00%, which is similar to the accuracy of the SVM. The LightGBM's complexity is represented by $O(d * n)$, where $d$ and $n$ represent the sample dimension and the number of samples, respectively. The LightGBM adopts a unilateral gradient algorithm to filter out the samples with low gradients in the training process, which reduces the calculation burden. It also is efficient in faster training. Compared with the SVM, its accuracy is slightly decreased because the number of data sets is insufficient, and a certain degree of accuracy is sacrificed for the training speed.

The classifier algorithms, SVM, LightGBM, and KNN, have different implementation scenarios. For example, the SVM and LightGBM are suitable for linearly separable data or fit nearly linearly separable data well, as well as problems that can be converted to linearly separable by kernel techniques. On the other hand, the KNN is suitable for small data sets and simple class decision boundaries. Therefore, more suitable classifiers are chosen based on the data characteristics.

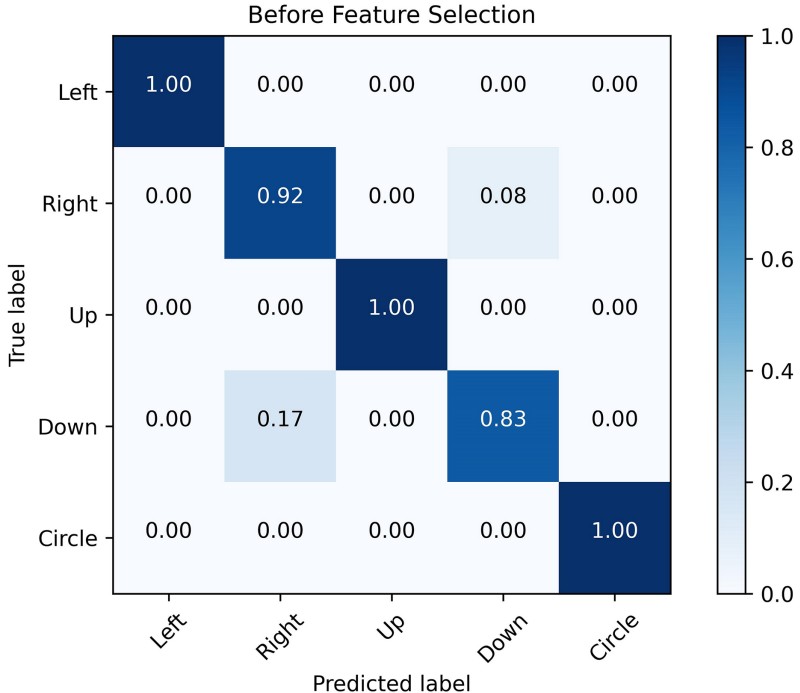

**Figure 12 Confusion matrix before feature selection.** The confusion matrix obtained by SVM classification algorithm before and after feature selection. It is analyzed from the figure that the misrecognition is mainly caused by two gestures, "right" and "down", which are caused by certain similarities in the waveforms of these two actions.

Therefore, the SVM is suitable for scenarios that require high reliability and are relatively fixed data, such as the fixed setting of a dedicated sensing device in the same scenario. At the same time, the LightGBM is suitable for scenarios that require a fast application and good recognition effect, and it can train an available and reliable model in time.

***The examination of the performance analysis before and after the feature selection process***

Compared with the algorithm not applying the feature selection procedure, the recognition accuracy of the conventional feature extraction procedure in the time domain is somewhat improved. The feature selection can effectively improve the robustness and generalization of the algorithm.

Figures 12 and 13 show the confusion matrix generated by the SVM before and after the feature selection process. The misrecognition is mainly caused by two gestures, "right" and "down", since they share certain similarities in the waveforms.

The proposed detection algorithm fits indoor and outdoor environments. However, the multipath effect of the indoor environment is more prominent, and indoor environment interference is less. Therefore, the accuracy of recognition in an indoor environment will be better.

Most researchers employ gestures to realize intelligent non-contact gesture control systems, which need motion-sensing devices similar to Kinect to optimize the real-time

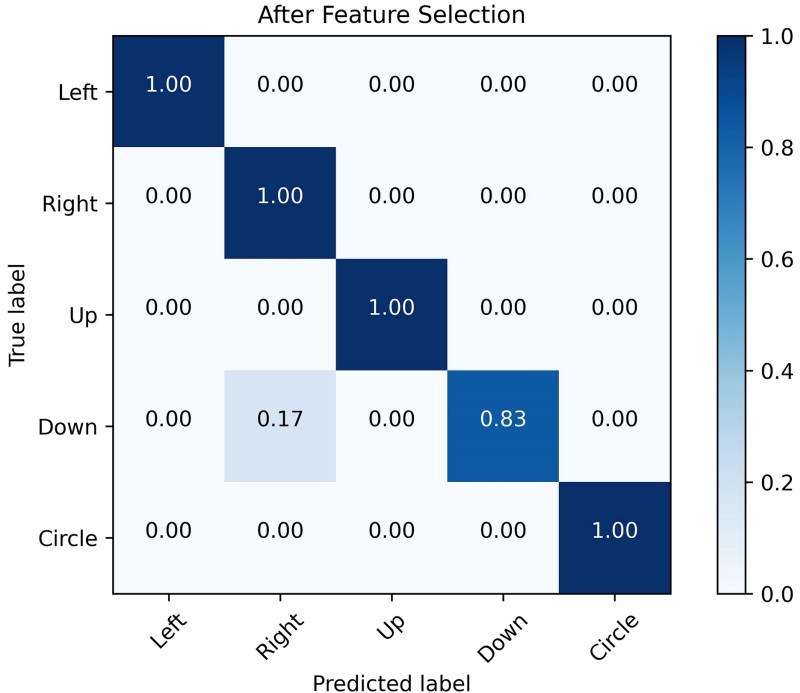

**Figure 13 Confusion matrix after feature selection.** The confusion matrix obtained by SVM classification algorithm before and after feature selection. It is analyzed from the figure that the misrecognition is mainly caused by two gestures, "right" and "down", which are caused by certain similarities in the waveforms of these two actions.               

image processing algorithm. The Kinect camera cover is implemented to investigate the specific movement changes of the human body and finally realize the control of smart homes according to human movements. However, the gesture control system based on Kinect is accompanied by the data leakage of users' private information and the inconvenience of wearing sensors. In contrast, the gesture control system implements Wi-Fi signals to recognize gesture actions to control smart homes without collecting any private information from users, providing a new idea for the research to promote the development of "smart homes".

## Future work
The conceptual model and experimental test are conducted in a small controllable space in the manuscript. The following problems need to be further researched.

### Multi-person interference
Other people passing by in the perceptual area will cause additional interference to the CSI dynamics collected by the receiver when subjects' gestures are studied. Thus, gesture recognition is poorly affected. For the proposed perception system, static objects in a sensing region will not affect the gesture information contained in CSI. However, for other people not related to the subject in the sensing area in the real-world scene, the generated interference signals can also be mixed into the CSI dynamics collected by the receiver.

Because of the intrinsic issues with Wi-Fi devices, multi-person gesture recognition will still be challenging.

### Lightweight model and device deployment

The experimental equipment consists of two notebooks equipped with wireless network cards. In future practical applications, miniaturized devices such as smartphones and wearable devices can be designed since they are easy to deploy and implement. Meanwhile, signals from multiple sources, such as Bluetooth, can be employed to enhance recognition accuracy. To consider using a lightweight algorithm to fit resource-limited end devices, a model compression technology is planned to be employed to decrease the number of parameters and enable real-time inference on terminal devices in the future.

### Real-time data learning

In the cross-domain scenario, the commonality of the pre-trained model will be reduced, and the model needs to update its parameters according to the current CSI data to adapt to gesture recognition in distinct environments and improve recognition accuracy. Therefore, a real-time online learning algorithm can be developed in the future so that it can be continuously fed from the real-time data stream and the algorithm parameters can be constantly updated according to the specific application scenarios.

## CONCLUSIONS

A system utilizing a Wi-Fi-sensing gesture control for smart homes is designed by theoretically investigating the Fresnel region sensing model to develop technical requirements of non-contact gesture control for family usage when transmitted and received signals are collected.

First, channel state information (CSI) related to gestures is collected in a specific area using transmitted and received Wi-Fi signals from wireless network cards. Non-invasive gesture recognition based on Wi-Fi signals presents an intriguing approach to achieving contactless gesture control in smart homes. To meet the technical requirements of non-invasive gesture recognition, a model based on the Fresnel zone is developed, utilizing the CSI data acquisition module of IEEE 802.11n wireless network card for experimental and theoretical analysis.

An improved dynamic double-threshold gesture interception algorithm is proposed to effectively extract relevant feature information from collected Wi-Fi signals while maintaining the integrity of gesture interception. An enhanced semi-supervised GAN is also introduced to achieve high-performance cross-scene gesture recognition with minimal labelled data, resulting in higher cross-validation accuracy even with a small number of collected samples.

The results demonstrate that the proposed dynamic double threshold gesture interception algorithm's accuracy can reach 98.20%, and the mean gesture recognition accuracy in a new environment can reach 95.67%, effectively identifying five different types of gestures and enabling contactless control for smart homes by utilizing wireless signals without compromising personal privacy or facing limitations associated with conventional visual and wearable device recognition systems.

The future direction of the research should contain multi-person gesture recognition due to the intrinsic issues with Wi-Fi devices, the implementation of a lightweight algorithm to fit resource-limited end devices, the decrease of the number of parameters, and the usage of real-time inference on terminal devices, the capability of the real-time data stream and constantly updated parameters of the algorithm in specific application scenarios.

### Funding
This study was supported by the Special Zone Project of National Defense Innovation. The funders had no role in study design, data collection and analysis, decision to publish, or preparation of the manuscript.

### Grant Disclosures
The following grant information was disclosed by the authors:
Special Zone Project of National Defense Innovation.

### Competing Interests
The authors declare that they have no competing interests.

### Author Contributions
- Chao Wang conceived and designed the experiments, prepared figures and/or tables, and approved the final draft.
- Yinfan Ding performed the experiments, authored or reviewed drafts of the article, and approved the final draft.
- Meng Zhou analyzed the data, authored or reviewed drafts of the article, and approved the final draft.
- Lin Tang performed the computation work, prepared figures and/or tables, and approved the final draft.

### Data Availability
The raw data are available in the Supplemental Files.

### Supplemental Information
Supplemental information for this article can be found online at http://dx.doi.org/10.7717/peerj-cs.2402#supplemental-information.

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
