# Peer review of "Wi-Fi sensing gesture control algorithm based on semi-supervised generative adversarial network"

_PeerJ Computer Science, doi:10.7717/peerj-cs.2402_

## Round 0.1 · original submission · Major Revisions

Dear authors,

Thank you for your submission, Your paper has been reviewed with interest by several experts in the field and you will see that they are advising against the publication in its current form. Their opinion and suggestions are valid and I do agree with them. Therefore, please carefully revise it along with the suggestions below

Editor Comments:

Please give more explanation of the Fresnel region sensing model, Gram Angle and Field (GASF) transform, and the enhanced generative adversarial network (GAN).
Please carefully elaborate on the details of the experimental setup, such as the number of participants, types of gestures tested, and environmental conditions, etc.
Please explain about rationale behind the choice of the GASF transform method
It would be nice if you could elaborate more on practical considerations, such as the system's robustness in different home environments (e.g., varying Wi-Fi signal strength, interference from other devices)
How about the scalability of the system when applied to larger and more diverse environments?
The conclusion should clearly summarize the main contributions and highlight the practical significance of the proposed system.
the language of the manuscript should be professionally done in revision.

·

Basic reporting

Paper has severe formatting and presentation issues. It is more like a engineering report rather research. The motivation for using GAN is also missing.

Experimental design

There are major flows in experimental section. It is almost impossible to re-implement the model while reading this paper. It is very hard to follow the steps.

Validity of the findings

There are no findings.

Additional comments

Paper is more engineering rather research. The paper lack basic writing and reporting skills. The figures are not providing any useful information, and also of very low quality.

Reviewer 2 ·

Basic reporting

Should be revised

Experimental design

The paper introduces a novel approach to non-contact gesture recognition for smart homes using Wi-Fi signals. It leverages a Fresnel zone-based sensing model and an enhanced dynamic double threshold algorithm for accurate gesture detection. The core innovation is a semi-supervised GAN that excels in cross-domain gesture recognition, even with limited labeled data.

Validity of the findings

Experimental results demonstrate high accuracy, significantly outperforming traditional methods like LDA and SVM. While the system shows promise, future work is needed to address challenges like multi-person interference, model scalability, and real-world deployment.

Additional comments

Comments to the Editor:
Detailed Comments: This paper has potential, particularly in its innovative approach to non-contact gesture recognition using Wi-Fi sensing technology. Still, the current quality needs to meet the relative requirements of this journal. Major revision should be done for this version of the paper strictly according to the following suggestions, and then it should be re-submitted to this journal. Therefore, I recommend a major revision. The shortcomings are as follows:
Shortcomings:
1. The paper needs to clearly identify the research gap in the existing literature. This would make it easier to understand the necessity and relevance of the proposed approach. Determining this gap would better justify the study's contribution.
2. The paper needs to compare the proposed model with other state-of-the-art techniques used for similar gesture recognition problems. This lack of comparison makes it difficult to assess the novelty and efficiency of the proposed approach.
3. The paper needs to provide a detailed analysis of how the proposed system can be scaled for deployment on resource-constrained devices like smartphones or IoT devices. It also does not adequately address computational complexity and energy efficiency.
4. The problem of interference from multiple users in the sensing area is mentioned but requires more thorough discussion. The paper does not include potential strategies for mitigating this interference, which is essential for practical applications.
5. The paper does not highlight the possibilities of applying data augmentation strategies to enhance the performance of the proposed model due to limited labeled data.
6. The paper does not include a quantitative analysis of the effect of the change in user behavior, like different speeds of gesture execution or variation in the size and shape of gestures.
7. Even though the paper presented is devoted entirely to smart home applications, it does not include a broader discussion of the approach's potential applicability and possibilities for using gesture recognition in healthcare or industrial automation, where it could be valuable.
8. The semi-supervised GAN model mentioned seems quite complicated to use in real implementation, and it could be a problem when implemented in low-power devices. This trade-off between complexity and practicality has yet to be thoroughly investigated.

Reviewer 3 ·

Basic reporting

The article ignores possible problems with network security, user privacy, and deployment simplicity and doesn't address how the system connects with current smart home technologies.

While the research emphasizes privacy as an advantage over camera-based systems, it ignores other ethical issues, such as the improper use of gesture recognition data. To create more obvious ethical barriers, the writers should go into further detail on the potential for unauthorized access to data and the repercussions of false recognition.

A thorough conclusion might summarise the results, discuss the study's ramifications, and possible uses outside of smart homes, and discuss how the methodology might be modified or expanded in further research.

Current references must provide information for the paper. To further strengthen the research, the writers are recommended to include more relevant and up-to-date literature.

The paper's language and grammar need significant improvements. Check the manuscript carefully and make any necessary edits as a result.

Experimental design

The 5GHz Wi-Fi band is used for all trials; however, the performance of the suggested system in other Wi-Fi frequencies is not considered, which could be a drawback.

Validity of the findings

It is important to remember that the most widely used metric for assessing a model's performance is accuracy. But there are other important factors to consider, like latency, computing complexity, and energy efficiency, all of which must be sufficiently taken into account.

The lack of discussion of the hardware prerequisites in the study could prevent the suggested system from operating, which could restrict the system's implementation.

One crucial topic for real-world applications is the stability of the suggested system in a dynamic environment, where conditions can vary over time. This is not covered in the paper.

It is challenging to evaluate the trade-offs because the authors have not yet compared the computational expenses of the suggested GAN model to those of the simpler models, such LDA or SVM.

While noise reduction is mentioned in passing in the report, more research is necessary to determine the potential effects of varying degrees of interference from other Wi-Fi devices on system performance.

---

## Round 0.2 · accepted · Accept

Based on the feedback received from the reviewer's in your revised manuscript, I'm pleased to inform you that your manuscript has been recommended for publication in it's current form. Thanks for your fine contribution

Reviewer 2 ·

Basic reporting

This paper now has a coherent structure and is written well, making it easy to follow. The authors have responded well to the research gap while emphasizing the significance of the proposed study. The additional information improves the comparison with previous approaches and contributes to the survey. Even though the technical terms used in the article are relevant, some of these terms could be elaborated for readers who are novices in the subject. Most of the results are easy to follow, and using figures and tables helps interpret the results.

Experimental design

It has been seen that the experimental design is enhanced with the help of comparisons with LDA and SVM, proving the model's effectiveness. New tests on scalability for smart handsets and IoT devices have been added to the concern of real-world deployment. Multi-user interference is now defined this requires further testing. This is also missing from the paper, yet it has factors that should be considered when deploying the model to resource-constrained devices.

Validity of the findings

The findings are accurate and consistent with further experiments conducted on other datasets and cross-domain testing. Further experiments for gesture variation, including speed and shape, are consistent with the proposed model. Data augmentation has been applied to enhance the model’s performance when not much-labeled data is available. The authors also have highlighted that increasing the size of the model leads to increased difficulties in deploying it on low-power devices, which is a major aspect of practical implementation.

Additional comments

The revised paper has added the possibility of applying the proposed solution in healthcare and industrial automation outside the smart home domain. Countermeasures to multi-user interference have been provided, but research is still needed to support these claims. Thus, the paper has been improved greatly and is ready for acceptance after these changes.

Reviewer 3 ·

Basic reporting

I am pleased to note that the authors have carefully considered the comments and suggestions raised in the previous review. The revisions made have significantly improved the clarity and comprehensiveness of the manuscript, successfully addressing all of my previous concerns.

I am particularly impressed with the quality of the technical writing and the overall presentation of the data. The manuscript is now well-structured and easy to follow

Experimental design

I am pleased to note that the authors have carefully considered the comments and suggestions raised in the previous review. The revisions made have significantly improved the clarity and comprehensiveness of the manuscript, successfully addressing all of my previous concerns.

I am particularly impressed with the quality of the technical writing and the overall presentation of the data. The manuscript is now well-structured and easy to follow

Validity of the findings

I am pleased to note that the authors have carefully considered the comments and suggestions raised in the previous review. The revisions made have significantly improved the clarity and comprehensiveness of the manuscript, successfully addressing all of my previous concerns.

I am particularly impressed with the quality of the technical writing and the overall presentation of the data. The manuscript is now well-structured and easy to follow